# Influence of Laser Parameters on the Texturing of 420 Stainless Steel

**DOI:** 10.3390/ma15248979

**Published:** 2022-12-15

**Authors:** Ângela Cunha, Flávio Bartolomeu, Filipe Silva, Bruno Trindade, Óscar Carvalho

**Affiliations:** 1CMEMS—Center for Microelectromechanical Systems, University of Minho, 4800-058 Guimarães, Portugal; 2LABBELS—Associate Laboratory, Braga/Guimarães, Portugal; 3CEMMPRE—Center for Mechanical Engineering, Materials and Processes, University of Coimbra, 3030-788 Coimbra, Portugal

**Keywords:** 420 stainless steel, laser surface texturing, parameters, microstructure, hardness

## Abstract

AISI 420 martensitic stainless steel is widely used in the mould industry due to its high tensile strength, hardness, and corrosion properties. Another requirement concerning any material used for this type of application is high thermal conductivity to minimise the time between consecutive injection cycles. The surfaces of some parts of the mould may be textured and reinforced with a material with higher thermal conductivity to achieve this aim. The results of a detailed study on the texturing of annealed 420 stainless steel using a Nd:YVO_4_ fibre laser are presented in this work. The influence of the laser’s processing parameters (laser power, scanning speed, number of passes, and line spacing) on the dimensions of the track, microstructure, and hardness of the modified surfaces was studied. Based on the continuity and dimensions of the machined grooves, several promising textures could be produced with laser power values from 5 to 30 W, scanning speeds of 500 to 2000 mm/s, 8 passes or more, and line spacings of 40 and 50 µm. High laser powers were responsible for the dissolution of chromium carbides in the laser tracks, the incorporation of chromium in austenite, and the consequent hardening of the microstructure.

## 1. Introduction

AISI 420 martensitic stainless steel (420SS) is a material widely used in the mould industry as well as other tooling applications, such as surgical tools [1,2,3], due to its high tensile strength, hardness, and corrosion properties [3,4,5,6]. In the annealed state, its structure consists of a ferritic matrix with a homogeneous dispersion of M_23_C_6_ spheroidised carbides with high ductility [7]. The tensile strength and hardness properties are improved by a quenching heat treatment, in which a martensite structure is formed [7]. The presence of alloyed chromium increases its resistance to corrosion [6,8]. However, this material has some limitations when good tribological properties are required, i.e., where controlled friction and increased wear resistance and lubrication are needed.

Nowadays, industries increasingly demand materials that meet the increasingly challenging demands of markets. For this reason, different approaches have been used to modify the surface properties of steel components, such as sandblasting [9], chemical etching [10], the deposition of coatings [11], electrode discharge machining (EDM) [10], electrochemical machining (ECM) [10,12], and texturing by electron beam, electric arc, or laser ablation [10,13,14], thus creating new physical, chemical, morphological, and metallurgical surface properties. The literature has reported that the required final properties of 420 stainless steel parts can be enhanced by surface modification using a laser. Steyer et al. [15] reported that surface melting by a laser is often used to improve the localised corrosion resistance inherent in stainless steels, namely, intergranular corrosion and pitting corrosion. Moreover, the hardening of metals using lasers has become an alternative approach to conventional methods, since it is a selective and localised heat source, limiting the expansion and distortion of the parts. For this reason, Netprasert et al. [16] studied the influence of laser parameters, namely, the scan overlap and scanning speed, on the microhardness and depth of the hardened layer. Further, Mahmoudi et al. [17] used a pulsed Nd: YAG laser to harden the surface of 420 stainless steel and compared it with the results obtained for the same material in the annealed and conventionally hardened state. Moreover, the authors studied the effects of laser parameters on the hardness and depth of the hardened area, as well as the effect of the overlap on pitting corrosion resistance.

Laser surface texturing, in particular, has become one of the major focuses of research and studies due to the flexibility of laser sources, the relative ease of automation, high precision, the relatively low cost and maintenance, and the commercial availability of robust laser systems [18,19]. Surface texturing is a manufacturing technology used to create dimples or grooves on the material’s surface [20]. Therefore, it has been applied in different fields to obtain different functional surfaces, namely for aesthetic purposes, and to improve the thermal efficiency of components as well as their tribological behaviour [21,22,23]. The laser texturing of metallic surfaces has been performed and commercial products have been successfully trialled in the automotive industry [24,25,26]. As far as steels are concerned, their low thermal conductivity is a limiting factor in obtaining good-quality textures. However, the literature on the surface texturing of 420SS is scarce. Pan et al. [27] used a picosecond laser for the fabrication of a superhydrophobic antibacterial textured surface on 420 stainless steel plates with the aim of reducing bacterial adhesion and delaying the formation of biofilms.

There are several types of lasers available for surface texturing/machining and material treatment, such as the CO_2_ laser, Nd: YAG laser, fibre laser, and excimer laser with wavelengths of 10.64 µm, 1064 nm, 1064 nm, and 193–351 nm, respectively [12]. Their use for a specific application depends on the material to be textured since it influences the energy absorbed [28,29].

The objective of this study was to study in detail the influence of laser parameters on the texturing process of 420 stainless steel by means of a Nd: YVO_4_ fibre laser. The aim was to develop new multi-functional surfaces for thermal applications, consisting of textured 420 stainless steel reinforced with highly conductive materials. Four hundred textures from eighty different combinations of processing parameters (power, scanning speed, and number of passes) were produced on a 420 stainless steel surface in this work. The results will be presented and discussed.

## 2. Materials and Methods

Square samples (35 mm × 35 mm × 3 mm) of 420 stainless steel (420SS) with 0.22% C, 0.004% S, 0.015% P, 0.02% O, 0.09% N, 0.29% Mn, 13.5% Cr, and Fe balance (values from the supplier) were textured using a Nd: YVO_4_ fibre laser (Model XM-30D Fibre Laser Marking Machine) with a wavelength of 1064 nm, maximum power of 30 W, spot size of 10 µm, frequency of 50 kHz, and pulse width of 10 ns. The texturing process is presented in Figure 1. The processing parameters (laser and drawing parameters and laser energy fluence) used in this work are shown in Table 1. The textures were designed using a full factorial design for 2 factors at 4 levels (laser power and scanning speed) and 2 factors at 5 levels (number of passes and line spacing) (4^2^ × 5^2^ = 400 experiments).

The laser energy fluence (*F*), defined as the laser energy divided by the area (J/mm^2^), was calculated according to the equation:$$F=\frac{P}{s\times l}\times n$$
where *P* is the laser power in W, *s* is the scanning speed in mm/s, *n* is the number of passes/loops, and *l* is the line spacing/interspacing between adjacent lines in mm. The laser energy fluence value varied from ~0 to 7680 J/mm^2^.

The textures were analysed morphologically by scanning electron microscopy (SEM) (NanoSEM-FEI Nova 200 (FEG/SEM)) by means of image analysis software (Image J 1.53e). Three-dimensional profilometry (InfiniteFocus from Bruker alicona) was used to determine the width and depth of the grooves at the mid-track point. Prior to the observations, the samples were polished with silicon carbide abrasive papers with successive grades up to a 4000 mesh. The influence of the laser parameters on the microstructure and grain size of 420 stainless steel was assessed by optical metallography after etching the samples with Vilella’s reagent. X-ray diffraction (XRD) (PANalytical X’Pert PRO MPD with Co Kα radiation) was used for the structural analysis. The hardness of the samples was evaluated by means of Shimadzu HMV-2 equipment with a load of 10 gf and a dwell time of 15 s.

## 3. Results and Discussion

### 3.1. Raw Material

Using XRD and optical microscopy, it was found that the raw 420 stainless steel plate was formed by ferrite and spheroidised chromium carbides, characteristic of an annealed state (Figure 2).

The overall chemical composition of the steel, as assessed by EDS, is presented in Table 2, but the carbon value is not in line with the chemical composition indicated by the supplier. The carbon value is too high for this steel, which may be explained by the possible surface contamination of the sample and by the fact that the measurements of the light elements of EDS have an important associated error.

### 3.2. Texturing

Figure 3 shows the SEM images of the textured 420SS produced with different processing parameters. The quality of the textures was evaluated considering two factors: the continuity of the machined grooves and the dimensions of cavities produced. Textures with a continuous width and depth of tens of micrometres were considered promising, which could be reinforced with micrometric powder particles (between 10 and 40 µm) in the future. Fourteen different combinations of processing parameters were found to give rise to suitable textures for this purpose (marked by a solid line in Figure 3). Another fourteen conditions (marked by a dashed line) corresponding to low laser powers and a high scanning speed did not reveal any visible cavities using SEM cross-section analysis, probably due to the accumulation of material at the site where the supposed laser ablation occurs or the lack of removal capacity (low laser energy fluence) [30,31].

The definition and quality of the grooves depend on the processing parameters. Low laser powers (1 and 16%) combined with high scanning speeds (2000 and 5000 mm/s) and line spacings of 40 and 50 µm led to low-quality and non-uniform textures characterised by discrete machined areas (Figure 4a). This phenomenon is due to the low laser energy fluence obtained for these parameter combinations. Furthermore, for low laser power (1%) and mainly for a low number of passes, it is possible to observe the continuous action of the laser, but non-machined areas are also observed (Figure 4b). Medium laser power values (16 and 64%), scanning speeds of 500 to 2000 mm/s, 8 passes or more, and line spacings of 40 and 50 µm resulted in the greatest number of good-quality grooves (Figure 4c). In fact, the definition of the machined lines increased as the line spacing increased because of the reduced interaction between successive passes of the laser.

For the maximum laser power (100%), high scanning speeds and greater line spacings are required to produce suitable textures. In this case, the number of passes plays an important role: for n = 1, the tracks are shallow (without a defined depth), whilst for n > 8, excessive remelting of the base material occurs. The resulting high laser energy fluence leads to an exaggerated increase in the surface temperature, which causes the material around the area being irradiated to melt, therefore impeding its ablation. This phenomenon was also detected and intensified for high laser powers and low scanning speeds and led to some material accumulating inside the grooves. This is particularly true for a high number of passes (Figure 5).

In fact, the amount of laser energy absorbed by the material decreases significantly throughout the thickness and has a great influence on the outcome on the surface. Therefore, if the energy is high enough, the material is ablated, creating a textured surface. This may occur when combining medium to high laser power with a medium scanning speed, as visible in Figure 4 for *P* = 64%, *s* = 2000 mm/s, *n* = 8, and *l* = 40–50 µm. On the other hand, a non-defined and remelted surface occurs when the amount of energy is insufficient for ablation (for example: *P* = 1%, *s* = 500 mm/s, *n* = 1, and *l* = 20–30 µm) or is too high and causes the material to overheat (for example: *P* = 100%, *s* = 100 mm/s, and *n* = 128), respectively.

The morphology of the abovementioned fourteen promising textures was analysed by 3D profilometry (Figure 6). In Figure 6, it is possible to observe four case studies referring to each of the laser powers, which exemplify the significant effect on the quality of the tracks produced. The combination of different processing parameters determines the quality and shape of the cavities produced. Analysing the laser power of 1%, it can be observed that the lines do not have a well-defined shape, presenting a dotted effect. Therefore, low laser power combined with a low scanning speed is not capable of producing a well-defined and uniform cavity. Increasing the laser power to 16% increases the definition of the cavity produced. The lines have a greater definition, which proves the increased efficiency of the ablation process. Furthermore, it is possible to see that the use of low scanning speeds causes the accumulation of material at the end of the track as a result of the material being dragged along during the ablation process. On the other hand, increasing the laser power (64%) and scanning speed (2000 mm/s) simultaneously worsens the definition of the cavity produced and leads to the accumulation of material along and at the end of the cavity. By increasing the laser power to 100% and comparing cases P64 s2000 n8 and P100 s2000 n8, it can be concluded that increasing the laser power leads to better line definition and the production of a cleaner cavity. Furthermore, the combination of high laser power (100%) with a high scanning speed (5000 mm/s) creates a poorly defined and discontinuous cavity along its length. The corresponding width and depth values were determined in the middle of the textured grooves and are presented in Figure 7. From Figure 6 and Figure 7, it is possible to conclude that for low laser powers (1%) and scanning speeds (100 mm/s), increasing the number of passes (from 128 to 256) is no longer beneficial in increasing the width. Experiments on other materials with nanosecond laser assistance have shown that any increase in groove width with an increase in the number of passes is limited [32]. According to the authors, as the mechanism that removes material in this type of laser is melt ejection, the molten pool may be deposited inside the wall of the groove and on the edge of the ablated region, reducing the width of the cavity created. Regarding the depth, as expected, it tends to increase with the number of passes [18,32]. The increase in laser power has a significant impact on the width and depth (cases analysed: P64 s2000 n8 l50 and P100 s2000 n8 l50). The percentage difference verified in the case of the width is 20% (decreases with increasing laser power value), while in the case of the depth, it is 16% (increases with increasing laser power value). Ablation occurs above a threshold fluence, which depends on the absorption mechanism, the material’s properties (structural phases and morphology), and the laser parameters (mainly wavelength and pulse duration) [33]. Ezhilmaran et al. [32] have stated that the width of the groove increases with the increase in laser energy per unit length (laser power/scanning speed) due to an increase in the fraction in the Gaussian beam that exceeds the ablation threshold. However, for the parameters mentioned above, this was not verified. Therefore, it was concluded that the increase in laser power (while maintaining the other process parameters) led to the excessive heating of the surface and the accumulation of material inside the cavity, therefore reducing the width.

In the case of the scanning speed, two cases were analysed: the use of low scanning speeds (100 and 500 mm/s) (combined with low laser power values) and the use of high scanning speeds (2000 and 5000 mm/s) (combined with high laser power values). In the former, the increase in scanning speed led to a sharp decrease in width, while in the latter, the variation in width was the inverse. On the other hand, in the case of depth, a low scanning speed had no significant impact, while a high scanning speed increased this value. In general, increasing the scanning speed leads to a reduction in the removal of the surface material since the number of pulses generated on the unit length decreases, and, consequently, there is a reduction in the pulse overlap [34,35]. However, an increase in scanning speed, when associated with the use of high laser power values, makes it possible to reduce this negative effect on the width of the cavity since the laser energy per unit of length is compensated by the high value of the laser power. Therefore, and analysing the cases presented in Figure 6, for low laser powers (16%) and scanning speeds (100 and 500 mm/s), increasing the scanning speed has a strong negative impact on the width and does not influence the depth, whereas for high laser powers (100%) and scanning speeds (2000 and 5000 mm/s), increasing the scanning speed has a strong positive impact on the width and depth. The line spacing proved not to be a determining factor regarding the width and depth of the textured grooves for the parameters studied.

Figure 8 shows the SEM images of the cross-sections of the grooves. For low laser powers (1 and 16%), the grooves produced present a waveform, with no noticeable differences when the other parameters (scanning speed, number of passes, and line spacing) are changed. On the other hand, when the laser power is increased (64 and 100%), the cavity has a more defined shape. In this case, the increase in the number of passes led to the formation of a cavity with a more funnelled shape. Moreover, for the same laser power, number of passes, and line spacing, the increase in scanning speed led to the formation of a shallower cavity since the laser spends less time at each of the points and therefore has less capacity to remove material (comparison of the cases presented for 100% laser power, where the only process parameter that varies is the scanning speed). Therefore, it is possible to conclude that the width, depth, and shape of the cavities are influenced not only by the laser energy fluence but also by each individual process parameter. For the same laser energy fluence values, the change in the laser power and scanning speed was shown to significantly affect these cavities’ characteristics (width, depth, and shape).

When comparing the results obtained by 3D profilometry and SEM (Figure 7 and Figure 8, respectively), a discrepancy in the width and, mainly, the depth of the grooves is observed. This can be explained by Figure 9, which shows that the depth of the cavity along the groove is not constant and decreases along the direction of texturing. This is due to the excessive increase in the surface temperature during texturing with the formation of a greater amount of the liquid phase that impedes the penetration of the laser beam and the consequent ablation of the material.

The influence of the processing parameters on the microstructures of the different textured samples was assessed by optical metallography on their cross-sections. Figure 10 shows two typical examples of textures obtained with different processing parameters.

For low laser powers, the scanning speed did not influence the microstructure of the base material (Figure 10a), formed by ferrite + chromium carbides (see Figure 2). However, for higher laser power values and more than one pass, white regions with no carbides could be detected on the top and the edges of the machined tracks (Figure 10b. EDS analysis performed in these white regions and in regions farther from the machined tracks revealed different carbon and chromium contents, and the values obtained in the white regions were lower (0.9 and 1.5 wt.% for C and 7.5 and 14.1 wt.% for Cr, respectively). It is likely that the remelting of the base material occurred for high laser powers with the loss of carbon and chromium. The microhardness results obtained for these two different regions were 320 ± 11 and 255 ± 5 HV_0.01_, respectively. The latter value is In line with that Indicated for this steel In the annealed state, while the former is higher, possibly due to the transformation of austenite into low-carbon and chromium martensite.

## 4. Conclusions

This study has shown that it is possible to successfully texture an annealed 420 stainless steel plate by using a Nd: YVO_4_ fibre laser. The range of processing parameters analysed had an influence on the ablation process. In general, medium to high values of laser power, medium scanning speeds, a high number of passes, and greater line spacings were required to obtain a good-quality texture. Therefore, the most promising textures were obtained with laser power values of 16, 64, and 100%, scanning speeds of 500 to 2000 mm/s, 8 passes or more, and line spacings of 40 and 50 µm. On the other hand, discrete or insufficient machined areas were obtained for low laser power values (1–16%), high scanning speeds (5000 mm/s) and line spacings (40–50 µm), and a low/medium number of passes (1–32). Moreover, remelting occurred for high laser powers (64–100%), low scanning speeds (100–500 mm/s), and a high number of passes (>8). The textures produced by high laser power values were responsible for microstructural changes on the top and edges of the regions adjacent to the machined ones (chromium carbide dissolution), with an increase in hardness.

## Figures and Tables

**Figure 1 materials-15-08979-f001:**
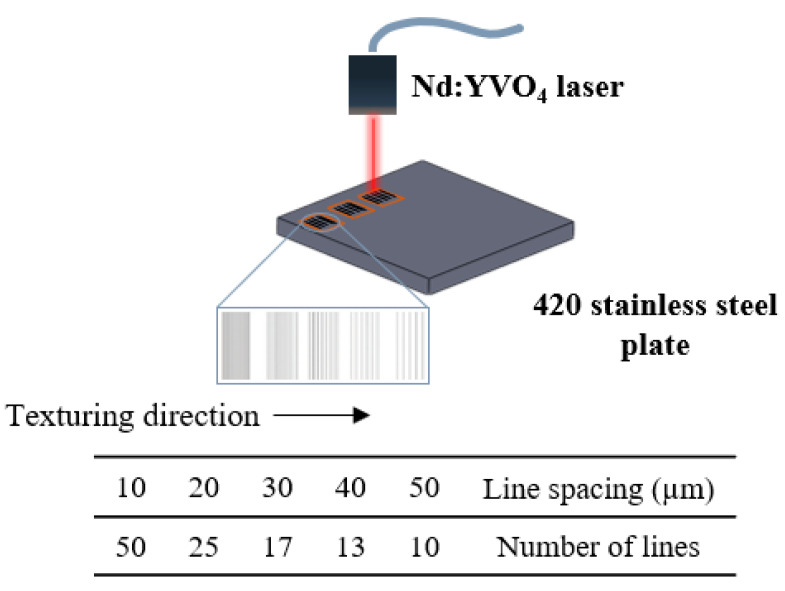
Texturing process: laser strategy with a different number of lines and line spacings.

**Figure 2 materials-15-08979-f002:**
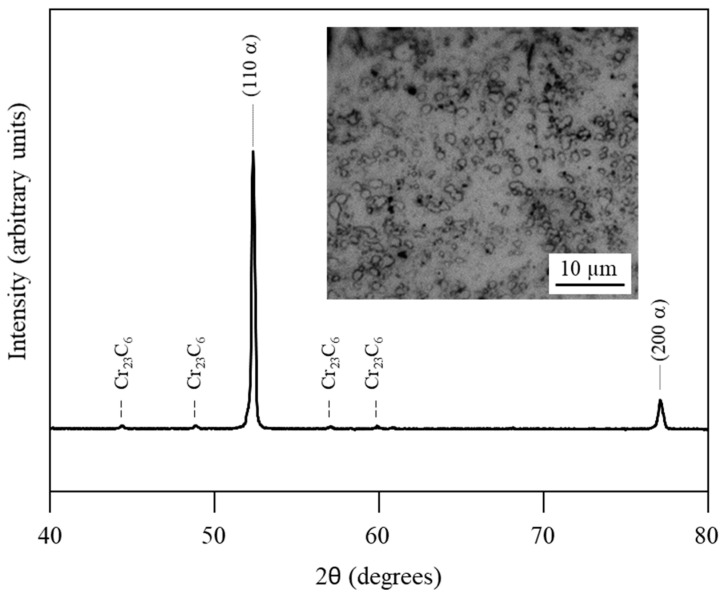
XRD pattern and optical micrograph of the raw 420 stainless steel plate.

**Figure 3 materials-15-08979-f003:**
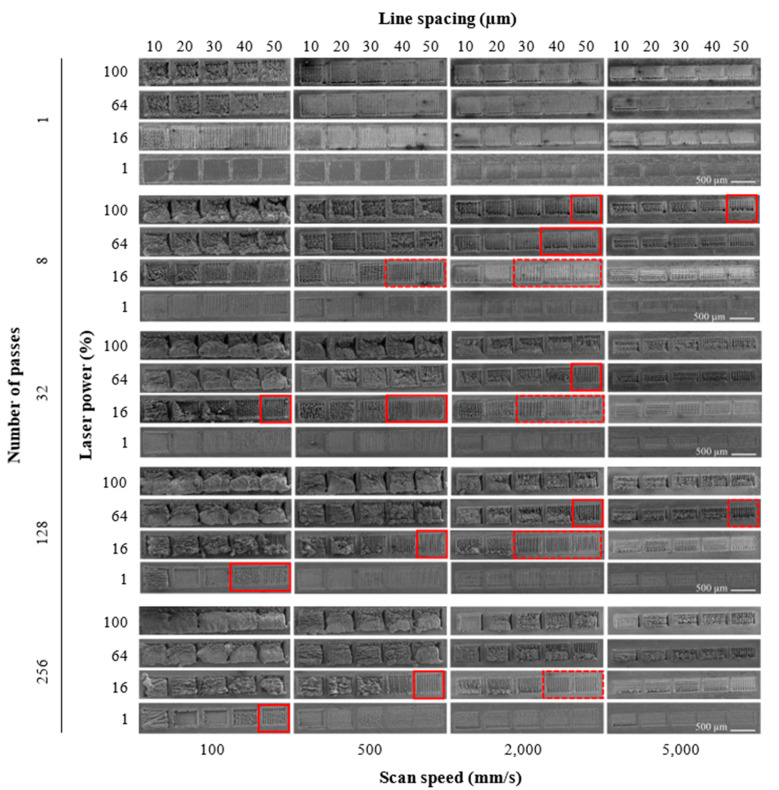
SEM images of textured 420 stainless steel samples subject to different processing parameters (laser power, scanning speed, number of passes, and line spacing).

**Figure 4 materials-15-08979-f004:**
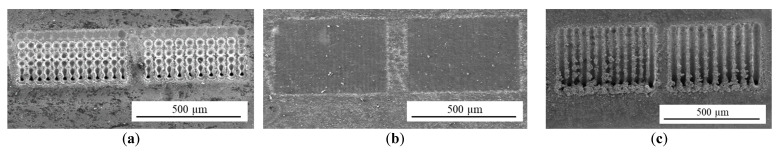
Examples of different phenomena that occur during texturing: (**a**) laser spot separation—discrete machined areas (*P* = 16%, *s* = 5000 mm/s, *n* = 8, and *l* = 40 and 50 µm), (**b**) insufficient machining (*P* = 1%, *s* = 500 mm/s, *n* = 1, and *l* = 20 and 30 µm), and (**c**) adequate machining (*P* = 64%, *s* = 2000 mm/s, *n* = 8, and *l* = 40 and 50 µm).

**Figure 5 materials-15-08979-f005:**
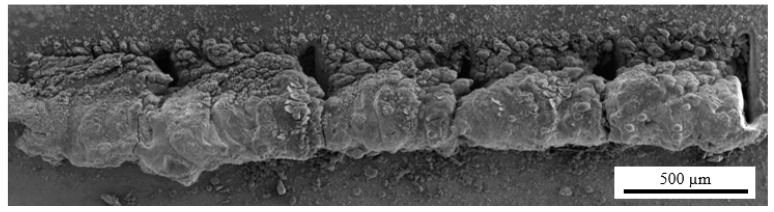
Example of the accumulation of material inside a groove when high laser energy is used (*P* = 100%, *s* = 100 mm/s, and *n* = 128).

**Figure 6 materials-15-08979-f006:**
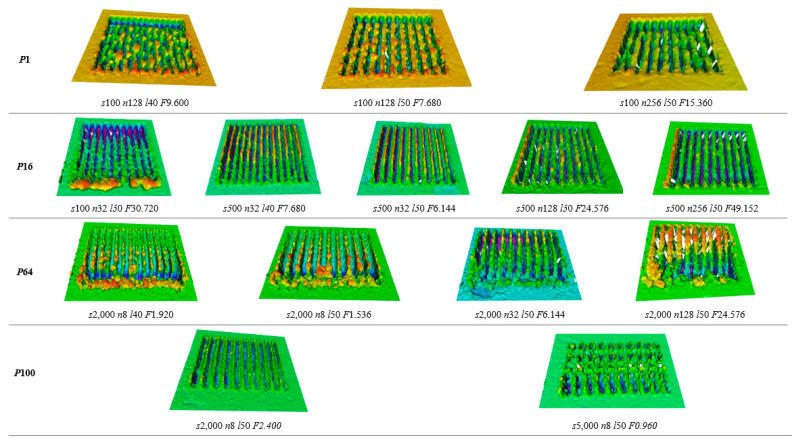
Three-dimensional optical profilometry images of the most promising textures (*P* = laser power in %, *s* = scanning speed in mm/s, *n* = number of passes, *l* = line spacing in µm, and *F* = laser energy fluence in J/mm^2^).

**Figure 7 materials-15-08979-f007:**
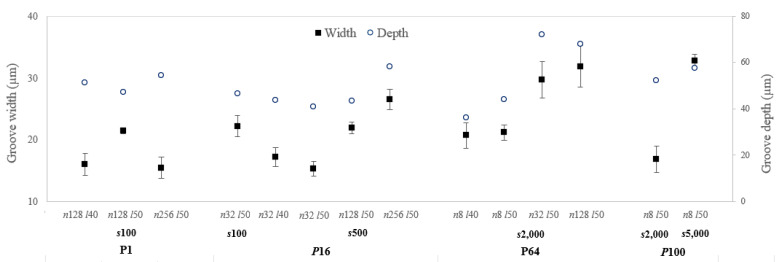
Width and depth in the middle of textured grooves obtained from different combinations of parameters (*P* = laser power in %, *s* = scanning speed in mm/s, *n* = number of passes, and *l* = line spacing in µm).

**Figure 8 materials-15-08979-f008:**
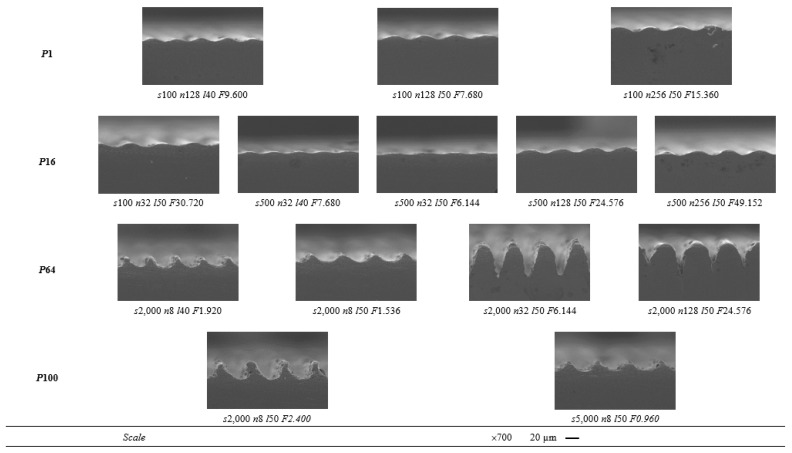
SEM images of the cross-sections of the fourteen samples with the most promising textures (*P* = laser power in %, *s* = scanning speed in mm/s, *n* = number of passes, *l* = line spacing in µm, and *F* = laser energy fluence in J/mm^2^).

**Figure 9 materials-15-08979-f009:**
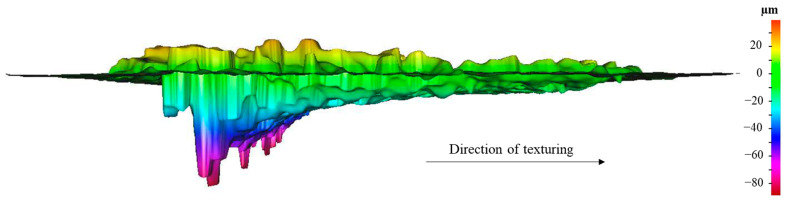
Depth profile of 420 stainless steel textured with *P* = 64%, *s* = 2000 mm/s, *n* = 8, and *l* = 40 µm.

**Figure 10 materials-15-08979-f010:**
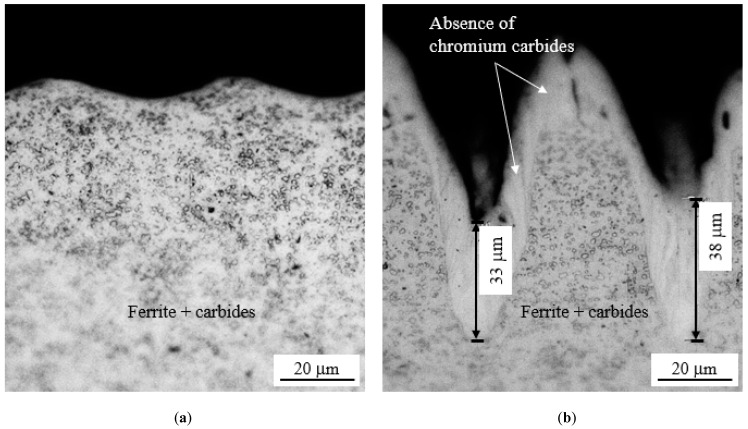
Typical examples of the microstructure of the 420 stainless steel after texturing using (**a**) low laser power and scanning speed (*P* = 16%, *s* = 500 mm/s, and *n* = 256) and (**b**) high laser power and scanning speed (*P* = 64%, *s* = 2000 mm/s, and *n* = 32). The line spacing in both cases was 50 µm.

**Table 1 materials-15-08979-t001:** Summary of the processing parameters used for the laser machining of the 420 stainless steel surface.

Laser Parameters			Drawing Parameters	Laser Energy Fluence—*F* (J/mm^2^)
Laser Power—*P* (%)	Scanning Speed—*s* (mm/s)	Number of Passes—*n*	Line Spacing—*l* (µm)	
1 (0.3 W)	100	1	10	
16 (4.8 W)	500	8	20	
64 (19.2 W)	2000	32	30	0 to 7680
100 (30 W)	5000	128	40	
		256	50	

**Table 2 materials-15-08979-t002:** Chemical composition of 420 stainless steel measured by EDS.

Element	Fe	Cr	Mn	C	P + S + O
(wt.%)	81.9 ± 0.5	14.1 ± 0.3	0.25 ± 0.08	1.5 ± 0.1	balance

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
