# Peer review of "Influence of Laser Parameters on the Texturing of 420 Stainless Steel"

_materials, 2022, doi:10.3390/ma15248979_

Round 1

Reviewer 1 Report

The manuscript entitled, “Influence of the laser parameters on the texturing of 420 stainless steel” studied the texturing process based on laser parameters of the 420 stainless steel. Though the results are interesting, the manuscript requires modifications.

1.      Page 2: Last sentence of the introduction “In this work, four hundred textures from eighty different combinations of processing parameters (power, scan speed, and number of passes) were produced on a 420 stainless steel surface. The results will be presented and discussed.”- Please mention the optimization technique or reference from which you select the sample size.

2.      Page 2: First sentence of Materials and methods “Square samples of 420SS (35 mm × 3 mm)”- Please attach the composition (EDS) of that alloy in the manuscript.

3.      Page 7: Mid-sentence of the paragraph “The ablation occurs above a threshold fluence, which depends on the absorption mechanism, the material’s properties (structural phases, and morphology), and the laser’s parameters (mainly wavelength, and pulse duration)”- Please elaborate the properties for which the change in ablation can possible.

4.      Page 8: Figure 5, a detailed explanation is required for a better understanding of the reader.

5.      Page 10: Last sentence of the paragraph “For the same laser energy fluence values, the change in laser power and scanning speed was shown to affect these cavities’ characteristics significantly (width, depth, and shape).” – Please discuss all the possibilities with all the changed parameters for the better understanding of the reader.

6.      Page 13: Figure 8 shows many texture variations for a single parameter during texturing process. Please explain the reason in detail for a better understanding of the reader.

7.      Page 14: First sentence of the paragraph “For low laser powers, the scanning speed did not influence the microstructure of the base material (Figure 9a), formed by ferrite + chromium carbides”- Please incorporate the EDS data in the manuscript.

8.      Page 14: The last sentence of Result and Discussion “The dissolution of the chromium carbides led to the incorporation of chromium in the austenite at a high temperature with the corresponding hardening of the microstructure after cooling effect”- Experimental data is required to validate the sentence. Please incorporate the data in the manuscript.

Round 2

Reviewer 1 Report

The manuscript may be accepted in the present form.